# The Impact of *IL1B rs1143634* and *DEFB1 rs11362* Variants on Periodontitis Risk in Phenylketonuria and Type 1 Diabetes Mellitus Patients in a Latvian Population

**DOI:** 10.3390/diagnostics14020192

**Published:** 2024-01-16

**Authors:** Darta Elizabete Emulina, Iveta Abola, Anda Brinkmane, Aleksejs Isakovs, Ingus Skadins, Georgijs Moisejevs, Linda Gailite, Madara Auzenbaha

**Affiliations:** 1Department of Conservative Dentistry and Oral Health, Riga Stradins University, LV-1007 Riga, Latvia; abola.iveta@gmail.com (I.A.); anda.brinkmane@rsu.lv (A.B.); 2Scientific Laboratory of Molecular Genetics, Riga Stradins University, LV-1007 Riga, Latviageorgijs.moisejevs@rsu.lv (G.M.); linda.gailite@rsu.lv (L.G.); madara.auzenbaha@rsu.lv (M.A.); 3Department of Biology and Microbiology, Riga Stradins University, LV-1007 Riga, Latvia; ingus.skadins@rsu.lv; 4Jekabpils Regional Hospital, LV-5201 Jekabpils, Latvia; 5Clinic of Medical Genetics and Prenatal Diagnostics, Children’s Clinical University Hospital, LV-1004 Riga, Latvia

**Keywords:** periodontitis risk, single nucleotide variants, phenylketonuria, type 1 diabetes mellitus, oral hygiene indices

## Abstract

Objectives: Periodontitis is a multifactorial disease that affects approximately 11% of the global population. The objective of this study was to examine whether, among individuals with phenylketonuria and type 1 diabetes mellitus, those with the *IL1B rs1143634* and/or *DEFB1 rs11362* genetic variants exhibit a higher periodontitis risk compared to healthy controls. Materials and Methods: In all, 43 phenylketonuria patients (aged 12–53), 28 type 1 diabetes mellitus patients (aged 11–40), and 63 healthy controls (aged 12–53) were included. The evaluation of periodontitis risk was conducted using the Silness–Löe plaque index, the Greene–Vermillion index, and an assessment for the necessity of calculus removal. Genetic variants *rs1143634* and *rs11362* were genotyped from salivary samples using restriction length polymorphism analysis. Results: The *DEFB1 rs11362* variant was associated with higher Silness–Löe and Greene–Vermillion index scores in phenylketonuria patients (*p* = 0.011 and *p* = 0.043, respectively). The *IL1B rs1143634* variant was associated with lower calculus removal necessity in type 1 diabetes mellitus patients (*p* = 0.030). Clinical examination showed the worst oral hygiene index scores for PKU patients. PKU patients also reported the least consistent tooth brushing and flossing habits. Conclusions: Genetic associations between *DEFB1 rs11362* and *IL1B rs1143634* variants and oral hygiene indices were observed in the PKU and T1DM groups, suggesting that genetic factors may contribute to periodontal health differences in these populations. Further research with a larger sample size is needed to confirm these findings and develop targeted oral health interventions.

## 1. Introduction

Periodontitis is a severe oral disease that is initiated by a build-up of dental plaque, resulting in infection, inflammation, and gradual degradation of the periodontium. The prevalence of severe periodontitis is estimated to be 743 million individuals worldwide, which accounts for approximately 11% of the global population [1]. Periodontitis can be attributed to both genetic and environmental etiological factors, whereby genetic polymorphisms associated with a higher production of pro-inflammatory factors may contribute to the severity of the disease [2].

Clinically, periodontitis is characterized by visual changes in the marginal gingiva, such as recession, redness, and swelling. Additionally, there is an increase in the depth of periodontal pockets, bleeding on probing, and deterioration of the dental supporting tissues (periodontal ligament and alveolar bone). As a result, teeth become more mobile and ultimately can be lost due to the irreversible destruction of the supporting bone [3]. The clinical manifestation of periodontitis varies depending on the patient’s age, the number and distribution of periodontal lesions, the severity of the lesions, and their location within the dental arch. Additionally, the level of oral biofilm contamination on the teeth also plays a role [4].

The diagnosis of periodontitis is based on the assessment of the disease manifestations through clinical and radiographical evaluations [5]. The initial obstacle in managing periodontal disease is achieving a prompt and precise diagnosis, which is particularly challenging due to the absence of pain in the early stages of the disease and patients’ infrequent pursuit of early treatment. Consequently, it is typical for periodontal disease to progress to advanced stages of severity before it is identified, and treatment is initiated [3]. In 2018, a new classification for defining periodontal disease was introduced. This system aims to provide a more accurate definition of periodontitis based on its stage, which considers the severity and extent of damage to the periodontal tissues as well as the complexity of treatment required. Additionally, the system also takes into account the grade of periodontitis, which indicates the risk of future progression [4,6]. The primary objective of the new classification system is to offer dental specialists a structured approach for personalized diagnosis, treatment, and prevention of periodontitis [6].

The pathogenesis of periodontitis involves numerous immune system molecules that cooperate to eradicate periodontal microorganisms while destroying periodontal tissues in the process [7]. The susceptibility and severity of periodontitis have been found to be associated with polymorphisms of genes involved in the immune response [7]. The gene polymorphisms of interleukin-1 (Il-1) are the most widely documented in studies related to the risk of periodontitis development [8,9,10,11]. The pro-inflammatory cytokine known as interleukin-1 beta (IL-1B) plays a significant role in the development of periodontitis by causing inflammation, bone resorption, and the breakdown of periodontal tissues [12]. *IL1B rs1143634* is the most reported genetic variation linked to an elevated susceptibility to the progression of periodontitis [13,14].

The expression of defensins is observed in multiple locations within the oral cavity, such as the periodontal tissues, oral mucosa, and salivary glands [15]. Defensins, because of their antibacterial and immunomodulatory characteristics, have the potential to contribute to the pathogenesis of various infectious and inflammatory conditions affecting the oral cavity, encompassing both soft and hard tissues [16]. It has been speculated that human beta-defensins may contribute to the progression of periodontitis, given the influence of *DEFB1* variants on peptide expression [17].

Phenylketonuria (PKU), a hereditary error of amino acid metabolism, is a disease caused by variants in the phenylalanine hydroxylase (PAH) gene. The absence of PAH enzymatic activity results in increased concentrations of phenylalanine in the bloodstream and the central nervous system. Untreated PKU has been associated with various adverse outcomes, including intellectual impairment, eczema, autism, seizures, and motor disabilities. To minimize the intake of phenylalanine, children diagnosed with PKU must adhere to a specialized low-protein dietary regimen. Previous research shows that, compared to healthy controls, PKU patients exhibit a greater likelihood of developing periodontal disease as they commonly lack the understanding and abilities necessary for good oral hygiene maintenance, and they are less likely to routinely visit a dentist or dental hygienist [18].

Type 1 diabetes mellitus (T1DM) is defined by insulin deficiency and ensuing hyperglycemia [19]. It is now understood that what was once thought to be a single autoimmune disorder caused by a T-cell-mediated attack on insulin-producing cells is actually the result of complex interactions between environmental factors, as well as differences in each patient’s microbiome, genome, metabolism, and immune system [19]. It is generally recognized that patients with all types of diabetes mellitus have a higher risk of developing periodontal disease than do those without diabetes [1,20].

The objective of this study was to examine whether, among individuals with PKU and T1DM, those with *IL1B rs1143634* and/or *DEFB1 rs11362* exhibit a higher susceptibility to the development of periodontal disease compared to healthy controls.

## 2. Materials and Methods

### 2.1. Ethics

Approval for this study was obtained from the Central Medical Ethics Committee (No. 1/19-03-26) and Genome Research Council of Latvia before the commencement of data collection. The study was performed according to the principles of the Declaration of Helsinki. Study objectives, possible risks, and participation benefits were explained to all participants or to the parents/legal guardians of underage participants prior to them signing an informed consent.

### 2.2. Study Subjects

The present study included 43 PKU patients who ranged in age from 12 to 53 years. Thirty-four individuals were diagnosed with PKU immediately after birth, while nine participants received a diagnosis at a later stage. Regarding treatment, only 2 out of the recruited 43 PKU patients underwent pharmaceutical treatment with synthetic tetrahydrobiopterin (BH4), while the remaining 41 patients managed their Phe levels solely through a restrictive diet. Only 1 of the PKU patients was overweight, while the other 42 did not have any associated metabolic disorders. Out of the 43 patients with PKU, 22 were completely adherent to their dietary restrictions, 8 patients partially adhered to their dietary needs, and 13 patients did not adhere to their diet at all.

The T1DM group consisted of a total of 28 patients aged 11 to 40 years. All recruited T1DM patients were diagnosed before the age of five. All patients with T1DM included in the study exhibited a well-controlled state of the disease, characterized by an average glucose level not exceeding 154 mg/dL (8.5 mmol/L), with an absence of proliferative diabetic neuropathy, absence of diabetic nephropathy (GFR < 60 mL/min) and absence of diabetic autonomic neuropathy. All enrolled T1DM patients adhered to their recommended diet (18–22 bread units, where one bread unit is equivalent to 12 g of carbohydrates).

The control group consisted of 63 healthy individuals who were matched with the T1DM and PKU groups in terms of age and gender. They were not following any specific diet, did not have any chronic diseases, and were regularly visiting the dentist for checkups.

### 2.3. Oral Health Assessment

Upon their arrival at the dental clinic, every participant or their representative completed a questionnaire pertaining to their dental hygiene practices. This questionnaire specifically inquired about the frequency of tooth brushing, as well as the utilization of dental floss and mouthwash.

The questionnaire was followed by salivary sample collection. Prior to their arrival at the dental clinic, all participants were provided with instructions to adhere to their regular breakfast routine and to engage in morning tooth brushing, ensuring that this activity was completed no later than 2 h before their scheduled appointment. Genomic DNA sample collection was performed using the unstimulated drain method to collect a salivary sample. The OG-500 (DNA Genotek, Stittsville, ON, Canada) Oragene saliva collection tube was provided to each patient for sample collection. Participants were instructed to assume a seated position with their heads slightly inclined downwards and to spit the saliva that was spontaneously secreted into the designated tube until it reached the indicated mark. The collection tube was then closed, gently mixed, and stored at room temperature until genomic DNA isolation was performed.

Clinical examination of the teeth and periodontal tissues was then performed by an individual dentist using dental magnifying loupes in an environment with sufficient and consistent lighting. The oral cavity was examined using a visuo-tactile approach, employing a blunt periodontal probe, dental mirror, and 3–1 syringe. The evaluation of oral hygiene status, gingival health, and periodontal disease risk was conducted using the Silness–Löe plaque index [21], the Greene–Vermillion index [22], and an assessment of the necessity of professional calculus removal. Both indices are thoroughly described in Appendix A. The determination regarding the requirement for professional calculus removal was made by a dental practitioner, based on the evaluation of the extent of hard calculus deposits observed on the teeth. See Figure 1 for photographs captured during clinical examination.

### 2.4. Genotyping

Genomic DNA was isolated from salivary samples by means of OG-500 (DNA Genotek, Stittsville, ON, Canada) Oragene using the protocol recommended by the manufacturer.

Genotyping of the selected single-nucleotide variations was conducted through the application of restriction length polymorphism analysis [23,24]; the methods are summarized in Table 1.

### 2.5. Statistical Analysis

The investigation included participants who had undergone an evaluation of their oral health and had successful genotyping results for the selected genetic variants. For the statistical analysis in this study, SPSS v29.0 (IBM Inc., New York, NY, USA) and Plink software v.1.9 [25] were used. The oral hygiene practices and oral hygiene indices were compared across all groups using the Kruskal–Wallis test and Fisher’s exact test; if a significant difference was observed, the study group (PKU and T1DM) was then compared to the control group by means of the chi-square test. The genetic associations between alleles, genotypes, and inheritance models for the minor allele were assessed in relation to clinical parameters. The results were considered to have statistical significance when *p* < 0.05.

## 3. Results

### 3.1. Oral Hygiene Assessment

In the assessment of oral hygiene practices, 9% of the PKU patients stated that they never brush their teeth, a response that was only encountered in this group. The T1DM and control groups showed higher adherence to tooth brushing, with 97% of the control group participants reporting regular brushing twice a day, which was the highest score for regular brushing among all three groups. The control group had the highest proportion of participants (65%) using dental floss. Although the use of mouthwash was not particularly popular in any group, the proportion of participants using mouthwash was highest in the PKU group (9%). The results regarding oral health hygiene practices are presented in Table 2 below.

In all, 93% of individuals in the PKU group required calculus removal, indicating the highest prevalence of calculus deposits among all three groups; this was followed by 68% among the control group individuals and 54% among the T1DM group patients. PKU patients exhibited the worst scores in the Silness–Löe and Greene–Vermillion indices. The percentage of individuals with the maximum score of three, implying more pronounced gingival inflammation, was by far the highest in the PKU group for both oral hygiene indices. Scores of two and one were distributed similarly across the groups, but the number of participants with a score of zero was highest in the control group for both indices. All results regarding the oral health hygiene index scores are presented in Table 3 and Figure 2 below.

### 3.2. Genetic Analysis Results

Firstly, the distributions of *IL1B rs1143634* and *DEFB1 rs11362* among patients diagnosed with PKU, those diagnosed with T1DM, and individuals from a healthy control group were compared. As anticipated, there was no observed correlation between any of the investigated variants and overall systemic health. Genotype distribution across all three study groups is presented in Figure 3 below. The comprehensive genetic analysis findings are presented in Appendix A. The following section presents the findings relating to the correlations between the allelic variants and oral hygiene indices across all three study groups.

### 3.3. DEFB1 rs11362

In the PKU group, when the minor allele (A) was suspected to be dominant, a statistically significant association with a higher Silness–Löe index score (*p* = 0.011) was observed. Homozygote or heterozygote genotypes were found in 93% and 92% of cases with scores of two or three, respectively. In contrast, such genotypes were present in 57% of cases with a score of one, while the presence of a score of zero was not observed in the PKU group. The Greene–Vermillion index is also linked to the minor allele (A) in a dominant manner (*p* = 0.043). The minor allele was observed in either homozygous or heterozygous state at frequencies of 100%, 75%, 70%, and 33% among those with index scores of three, two, one, and zero, respectively.

No associations between variant and oral health status were observed in the T1DM and control groups.

### 3.4. IL1B rs1143634

In the T1DM group, in cases where the minor allele T was inherited in a recessive way, an association with a requirement for calculus removal was identified (*p* = 0.030). The results revealed that 50% (4 out of 8 individuals) of individuals with the minor allele in a homozygote state did not require calculus removal, whereas, in the group that did require calculus removal, the corresponding proportion was 10% (2 out of 19 individuals).

No statistically significant associations were found in the PKU and control groups.

## 4. Discussion

Periodontitis is a multifactorial disease that can be influenced by a wide range of systemic conditions, the oral microbiome’s interaction with the host immune response, tobacco smoking, medications, age, gender, genetic predisposition, stress, and even factors such as socioeconomic status and education [26,27]. However, it is generally agreed upon that the presence of plaque is required to initiate an inflammatory response, which can later progress into periodontal destruction in susceptible patients [28]. Therefore, higher scores in oral hygiene indices such as Silness–Löe and Greene–Vermillion cannot be used as diagnostic tools for periodontal disease as they do not consider some important parameters for diagnosing periodontal disease (such as probing depths to measure bone loss, an assessment of tooth mobility, and furcation involvement), but they can be useful periodontitis risk indicators. Our results revealed that PKU patients exhibited the highest scores in both oral hygiene indices, signifying a higher level of gingival inflammation and plaque accumulation and, consequently, an elevated susceptibility to periodontal disease. These results show that PKU patients, who reported the worst tooth brushing and flossing habits out of all three study groups, subsequently had the worst gingival health clinically. Although the available literature on oral health differences in PKU patients is limited, some prior studies support our findings. For example, Lucas and colleagues found that PKU patients presented with more plaque than healthy controls [29]. Singh-Hüsgen et al. found that T1DM patients displayed slightly higher Silness–Löe index values compared to PKU patients and healthy controls [30], but it must be noted that this research only included children between the ages of 3 and 18 years. In our study, in addition to adults, individuals from the age of 11 years with all secondary teeth present were included. These findings underscore the importance of regular dental check-ups, oral hygiene instructions, and preventive dental care for PKU patients especially to minimize oral health complications.

The genetic analysis performed in the current study aimed to explore associations between specific allelic variants and oral hygiene indices across the study groups. For the *DEFB1 rs11362* variant, a statistically significant association was found between the minor allele (A) and a higher Silness–Löe index score. This suggests that individuals with this genetic variant may be more predisposed to gingival inflammation and plaque accumulation. Similarly, the Greene–Vermillion index was associated with the minor allele (A) in a dominant manner. These findings imply that this genetic variant may affect gingival health and plaque control. The *IL1B rs1143634* variant was proven significant with the T1DM group, particularly when the minor allele (T) was inherited recessively, implying that T1DM patients with the *IL1B rs1143634* variant required calculus removal less frequently. This suggests that individuals with T1DM and this genetic variant may have a decreased risk of calculus formation and, therefore, a decreased periodontitis risk.

Current research results on the associations of genetic variants with periodontitis severity are highly variable. Although we did not find any prior studies that looked at periodontitis risk in conjunction with oral hygiene indices and PKU/T1DM patients specifically, the role of genetic variants is a widely studied subject in a variety of populations and study designs. Regarding *DEFB1 rs11362* and periodontitis, the current research shows variable results. While some authors have been able to prove a role of *DEFB1 rs11362* in increased periodontal disease severity [17,31,32], other research did not find any such association [33], and some authors argue that *DEFB1 rs11362* is linked to an increased caries development risk, rather than periodontitis [34,35]. The inconsistency in these results could be due to the multifactorial nature of periodontitis, as it is very difficult to account for all etiological factors that could be influencing the disease severity in each study population. In our study, we included individuals with well-controlled T1DM. There are several examples in the existing research where the *IL1B rs1143634* variant was not proven to increase periodontal disease risk [36,37]. However, some authors have found a statistically significant association between *IL1B rs1143634* and periodontitis severity [8,14,38].

Nutrition constitutes another significant determinant that impacts oral health. All participants of the current study were questioned regarding their dietary habits. A crucial criterion for inclusion based on the dietary patterns of participants in the control group was the absence of a dietary regimen that excluded any significant food groups to better represent the general population and to counter the PKU and T1DM populations, who must follow specific diets. Control individuals adhering to specific dietary patterns such as veganism, vegetarianism, carbohydrate restriction, or a ketogenic diet were excluded from the study. The inclusion of participants adhering to an all-inclusive dietary regimen was deemed crucial to consider potential oral health changes that may arise from the exclusion of specific food groups. For example, omnivores seem to be more prone to dental caries and eventual loss of all teeth, and they also have worse periodontal health indices when compared to vegans and vegetarians, who, in turn, are more predisposed to dental erosion due to the highly acidic nature of their nutrition [39]. Atarbashi-Moghadam and colleagues came to similar conclusions as they found that raw vegans had better periodontal health indices when compared to controls, but just like Azzola and colleagues, they emphasized that vegan participants often exhibited better oral hygiene practices and overall healthier lifestyle choices [40]. Eberhard et al. found that following a semi-vegetarian high-fat diet improved periodontal health indices when compared to an omnivorous higher-fat or higher-carbohydrate diet or a semi-vegetarian higher-carbohydrate diet, implying that the consumption of less meat and carbohydrates might benefit periodontal health [41]. T1DM patients are generally advised to follow a healthy and balanced diet, just like systemically healthy people. However, one of the most important cornerstones in the management of their condition is limiting dietary sugars and carbohydrates to improve blood glucose levels [42].

The current study is one of a kind, as there are no other data regarding correlations between oral hygiene indices and genetic variants for PKU and T1DM patients in the context of their special dietary needs in the current literature. It must be pointed out that the current research included a relatively small study group, and to make concrete conclusions, a similar study should be carried out with more participants. Further research is warranted to elucidate the precise mechanisms underlying the genetic associations observed in this study and to develop targeted strategies for improving oral health outcomes in these populations.

## 5. Conclusions

The genetic associations found in the current study were specific to the PKU and T1DM groups and were not observed in the control group, emphasizing the potential genetic factors that may contribute to oral health differences among these populations in association with specific eating habits. These findings underscore the need for tailored oral health interventions for PKU and T1DM patients to address gingival health and plaque control.

## Figures and Tables

**Figure 1 diagnostics-14-00192-f001:**
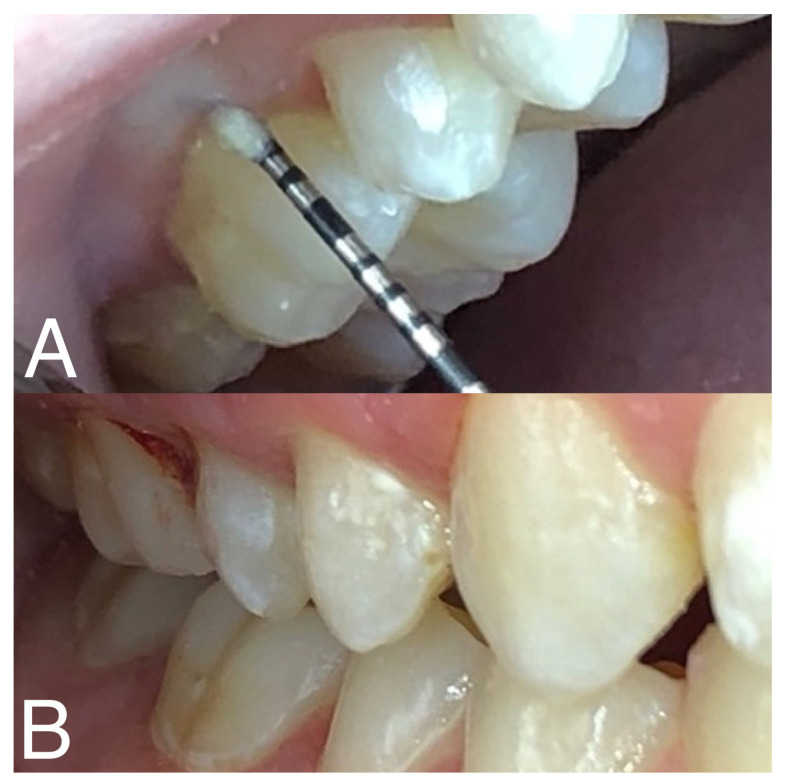
Photographs captured during clinical assessment of a phenylketonuria patient. Photograph (**A**) depicts the evaluation of plaque accumulation along the gum line using a periodontal probe. Photograph (**B**) exhibits minor bleeding of gingiva on the buccal and mesial surfaces of an upper right first molar following gentle probing during examination.

**Figure 2 diagnostics-14-00192-f002:**
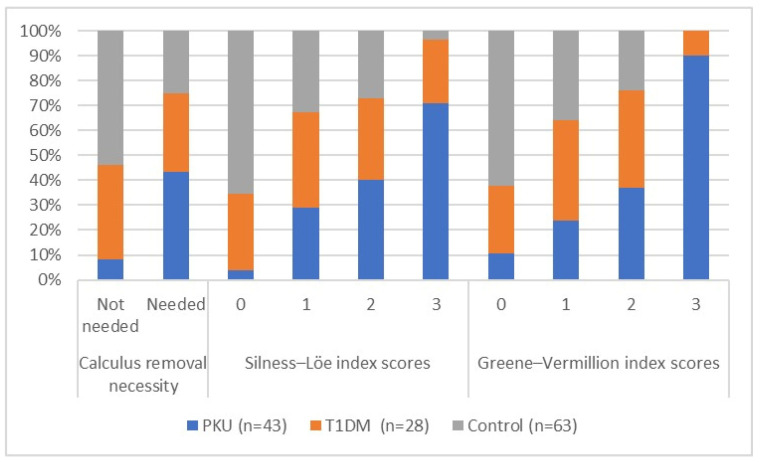
Visual representation of oral hygiene index scores and calculus removal necessity based on clinical examination in all study groups (data from Table 3).

**Figure 3 diagnostics-14-00192-f003:**
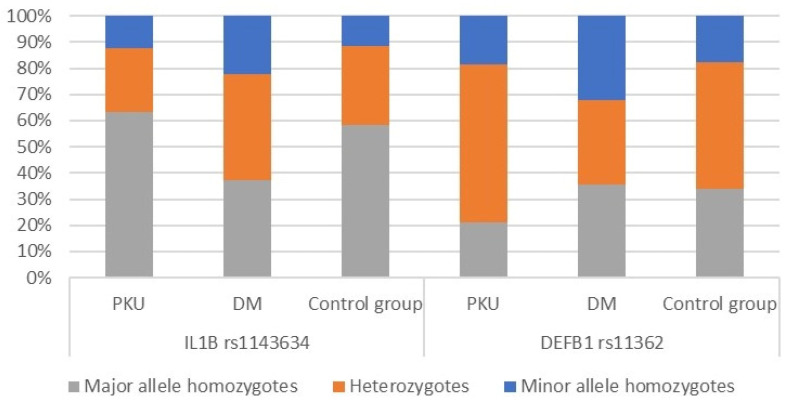
Genotype distribution across all three study groups.

**Table 1 diagnostics-14-00192-t001:** Description of genotyped genetic variants and used method.

Genetic Variant	HGVS, Minor Allele	Primer Sequences	Restriction Enzyme
*IL1B rs1143634*	NM_000576.3:c.315C > T p.(Phe105=), T (A *)	5′-GTTGTCATCAGACTTTGACC-3′ 5′-TTCAGTTCATATGGACCAGA-3′	*Taq1* (fragments for A allele 250 bp, for G allele 135 and 115 bp) [24]
*DEFB1 rs11362*	NM_005218.4:c.-20G > A p.(?), A (T *)	5′-CAGGGGTTAGCGATTAG-3′ 5′-GCAGAGAGTAAACAGAAGGTA-3′	*BstnI* (fragments for G allele 167 bp and 60 bp, for A allele 227 bp) [23]

HGVS—variant nomenclature according to Human Genome Variation * Allele name according to chromosomal position from the GnomAd database.

**Table 2 diagnostics-14-00192-t002:** Oral hygiene practices based on questionnaire results in all study groups.

	PKU Group (*n* = 43)		T1DM Group (*n* = 28)		Control Group (*n* = 63)	
Frequency of teeth brushing						
Do not brush	4	9%	0	0%	0	0%
Once per day	13	30%	12	43%	2	3%
Twice per day	26	60%	16	57%	61	97%
	*p* * < 0.001		*p* * < 0.001			
Use of supplementary oral hygiene items						
Do not use	33	77%	15	54%	21	33%
Dental floss	6	14%	11	39%	41	65%
Mouthwash	4	9%	2	7%	1	2%
	*p* * < 0.001		*p* * = 0.048			

* Statistical significance calculated between the case and control groups.

**Table 3 diagnostics-14-00192-t003:** Oral hygiene index scores and calculus removal necessity based on clinical examination in all study groups.

	PKU Group (*n* = 43)		T1DM Group (*n* = 28)		Control Group (*n* = 63)	
Calculus removal necessity						
Not needed	3	7%	9	32%	29	46%
Needed	40	93%	19	68%	34	54%
	*p* < 0.001 *		*p* = 0.156 *			
Silness–Löe index scores						
0	1	2%	5	18%	24	38%
1	14	33%	12	43%	23	37%
2	15	35%	8	29%	15	24%
3	13	30%	3	11%	1	2%
	*p* < 0.001 *		*p* = 0.088 *			
Greene–Vermillion index scores						
0	3	7%	5	18%	26	41%
1	10	23%	11	39%	22	35%
2	16	37%	11	39%	15	24%
3	14	33%	1	4%	0	0%
	*p* < 0.001 *		*p* = 0.069 *			

* Statistical significance calculated between the case and control groups.

## Data Availability

Data presented in this study are available on request from the corresponding author.

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
