# Peer review of "The Impact of IL1B rs1143634 and DEFB1 rs11362 Variants on Periodontitis Risk in Phenylketonuria and Type 1 Diabetes Mellitus Patients in a Latvian Population"

_diagnostics, 2024, doi:10.3390/diagnostics14020192_

Round 1

Reviewer 1 Report

Comments and Suggestions for Authors

This study investigated the potential association between genetic variants and periodontitis risk in individuals with phenylketonuria (PKU) and type 1 diabetes mellitus (T1DM) compared to healthy controls. The research focused on the IL1B rs1143634 and DEFB1 rs11362 variants and their impact on periodontal health. The results revealed that the DEFB1 rs11362 variant was linked to higher Silness–Löe and Greene–Vermillion index scores in PKU patients, indicating an increased risk of periodontitis (p = 0.011 and p = 0.043, respectively). Conversely, the IL1B rs1143634 variant was associated with a reduced necessity for calculus removal in type 1 diabetes mellitus patients (p = 0.030). Clinical examination highlighted poorer oral hygiene indices in PKU patients, who also reported less consistent tooth brushing and flossing habits. The findings suggest that genetic polymorphisms (DEFB1 rs11362 and IL1B rs1143634), may contribute to differences in periodontal health in individuals with phenylketonuria and type 1 diabetes mellitus when compared to healthy controls.

I recommend the improvement of the text considering these details:

 -              To write more about the PKU patients included into the ‘2.2. Study Subjects’; for PKU patients - if are in metabolic ranges (which were the blood Phenylalanine values); if the disease was detected at birth and followed the treatment from the first weeks of life. Were the Phe values obtained with  the restrictive diet/PAL (Palinzique) and/or BH4 treatment (if applicable)? Which were the values of their QI (were the patients respecting the diet?). If they associate other metabolic conditions or diseases, like overweight etc.

-              To give more details about T1DM participants regarding other comorbidities that could affect their inflammatory status.

-              To mention if the subject evaluated into the Figure 1 is belonging to the control group or patiemts. And in Fig. 1B: to indicate where is the ‘minor bleeding’.

Reviewer 2 Report

Comments and Suggestions for Authors

The clinical characterzation of perioodotal disease need to done - in addition as it has been done - also by the new periodontitis classification, see Sorsa etal Diagnostics-20, Räisänen et al Biomedicines-23.

Comments on the Quality of English Language

English should be revised a native english speaker.

Round 2

Reviewer 2 Report

Comments and Suggestions for Authors

Revision well done.

Accepted for publication.

Comments on the Quality of English Language

Same as above.